# The Conservation of Low Complexity Regions in Bacterial Proteins Depends on the Pathogenicity of the Strain and Subcellular Location of the Protein

**DOI:** 10.3390/genes12030451

**Published:** 2021-03-22

**Authors:** Pablo Mier, Miguel A. Andrade-Navarro

**Affiliations:** Institute of Organismic and Molecular Evolution, Faculty of Biology, Johannes Gutenberg University, 55128 Mainz, Germany; andrade@uni-mainz.de

**Keywords:** low complexity regions, homorepeats, compositionally biased regions, bacterial strains, orthology, host–pathogen interactions

## Abstract

Low complexity regions (LCRs) in proteins are characterized by amino acid frequencies that differ from the average. These regions evolve faster and tend to be less conserved between homologs than globular domains. They are not common in bacteria, as compared to their prevalence in eukaryotes. Studying their conservation could help provide hypotheses about their function. To obtain the appropriate evolutionary focus for this rapidly evolving feature, here we study the conservation of LCRs in bacterial strains and compare their high variability to the closeness of the strains. For this, we selected 20 taxonomically diverse bacterial species and obtained the completely sequenced proteomes of two strains per species. We calculated all orthologous pairs for each of the 20 strain pairs. Per orthologous pair, we computed the conservation of two types of LCRs: compositionally biased regions (CBRs) and homorepeats (polyX). Our results show that, in bacteria, Q-rich CBRs are the most conserved, while A-rich CBRs and polyA are the most variable. LCRs have generally higher conservation when comparing pathogenic strains. However, this result depends on protein subcellular location: LCRs accumulate in extracellular and outer membrane proteins, with conservation increased in the extracellular proteins of pathogens, and decreased for polyX in the outer membrane proteins of pathogens. We conclude that these dependencies support the functional importance of LCRs in host–pathogen interactions.

## 1. Introduction

Low complexity regions (LCRs) are protein regions with a skewed amino acid composition [1]. Assuming the expected general composition of a complete proteome as the background (with inherent biases, such as the higher proportion of amino acids leucine and serine, and the lower proportion of the aromatics tryptophan, tyrosine, and phenylalanine), LCRs diverge from it either towards a specific amino acid (e.g., alanine-rich region) or an amino acid type (e.g., highly hydrophobic).

Compositionally biased regions (CBRs) and homorepeats (polyX) are among the most common LCRs. Both rely on a higher proportion of one amino acid, either scattered (CBRs) or localized (polyX) [1]. They show a high mutation rate [2]; as a result, orthologous proteins can have very different LCRs. 

In addition to having high variability, LCRs have unusual structural properties, making their study more complex than that of globular domains. In eukaryotes, LCRs are increasingly found to be sites for protein post-translational modifications with regulatory effects on protein interactions [3,4,5]. LCRs are much less common in prokaryotes [6] and therefore this has cast doubt about whether they have any functional relevance in these organisms. However, the abundance of completely sequenced prokaryotic genomes enables the analysis of the conservation of protein features across homologs, which can be employed to provide hypotheses about LCR conservation and function in prokaryotes. This approach was used to provide evidence that the conservation of LCRs can be connected to general functions in prokaryotes [7]: translation, nucleic acid binding, metal-ion binding, and protein folding. 

In comparison to eukaryotes, bacteria are able to evolve much faster, also in the context of pathogenic bacteria under selective pressure from their interactions with their hosts. In particular, LCRs resulting in disordered regions such as adhesins are known to be determinants of pathogenicity [8]. Here, we focus our study on aspects of LCR function related to this short evolutionary scale. 

Completely sequenced bacterial genomes provide a wealth of homologs for comparison of LCR conservation, not only for different species but also for the finer taxonomic level defined by strains. While the term ‘species’ is the most basic taxonomic rank in biological classification [9], its definition varies between eukaryotes (individuals who can mate and produce fertile offspring) and prokaryotes (group of individuals who form a coherent genomic cluster [10,11]). The latter can be clearly problematic, and metagenomic studies have lately complicated the situation [12,13,14]. Mass genomic sequencing of prokaryotic colonies brings within-species diversity to a genomic similarity continuum between species [15,16]. We rely on the definitions used in the sequence databases indicating sequence provenance (see Materials and Methods for details). Even though the lines between species can be blurred at times, strains should provide more similar individuals (proteomes) within species to derive general rules for the evolution of LCRs in bacterial species with high confidence. 

In this work, we study how the variability of LCRs relates to the proteome similarity of bacterial strains. We compare the sequence similarity between bacterial strain pairs with the conservation of CBRs and homorepeats. We contrast the results with the conservation of protein domains. For simplicity, we consider a feature (CBR, homorepeat, domain) as conserved if it is present at overlapping positions between two aligned sequences (see Materials and Methods for details). The feature type has to be identical (enriched amino acid in the CBR, homorepeat amino acid, domain type). The conservation level of the feature is considered globally as the fraction of features (CBRs, homorepeats or domains) conserved.

In our analysis, we then consider the pathogenicity of the strains as a major feature towards the conservation of the LCRs. Here, we find that the subcellular location of the compared proteins is a decisive factor.

## 2. Materials and Methods

### 2.1. Data Retrieval

A total of 20 bacterial species from taxonomically diverse taxa were manually selected. For the selection process, the species must meet the following conditions: (1) species must have at least two completely sequenced strains (not subspecies), and (2) they must be taxonomically distant to the rest of the selected species. The proteomes of two strains per species were downloaded from the UniProtKB database v2020_06 [17] (Appendix A). Taxonomic information was obtained from the NCBI Common Taxonomy Tree resource [18].

### 2.2. Orthology Assessment

For each pair of strains, we calculated the complete set of orthologous pairs using ProteinOrtho v6.0.23 [19] with default parameters. We also obtained the bit score of similarity for each pair of orthologs. We used the average of the bit scores of similarity of all orthologous pairs as a measure of the sequence conservation between the proteomes of both strains. 

We did a similar orthology assessment for 190 more species pairs consisting of all possible pair combinations of one representative per strain pair.

### 2.3. Conservation of Protein Features

All protein sequences were analyzed locally with InterProScan version 5.47-82.0 (default parameters) [20], sQanner (requiring a minimum of four identical consecutive amino acids) [21], and PlaToLoCo (running CAST with default parameters) [22], to positionally annotate their domains, polyX, and CBRs, respectively. In the case of CAST, each sequence is compared against a database of 20 different homopolymers, one per amino acid, using BLAST. Regions found with higher bit scores than the threshold (40 bits by default) are detected as X-rich CBRs.

We evaluated whether particular sequence features (polyX, CBRs, and domains) were conserved between orthologs. A feature (e.g., a polyQ, an A-rich region, or an SH3 domain) is considered conserved in a pair of orthologs if both sequences have it in overlapping coordinates of their pairwise alignment, and at least 50% of the longest region overlaps with the predicted feature in its ortholog. We calculated a conservation ratio for each feature type in each pair of strains (or species) as the fraction of features that are conserved (e.g., a conservation ratio of one indicates that a feature is always conserved).

### 2.4. Subcellular Location Prediction

The subcellular location of each protein was predicted with PSORTb v3.0.2 [23]. For the prediction, the parameter Gram stain per proteome was mandatory; the value we used per strain can be found in Appendix A. The subcellular locations we considered are as follows: cytoplasmic membrane, cytoplasmic, extracellular, outer membrane, and periplasmic.

## 3. Results

### 3.1. Conservation of LCRs in Bacterial Strains

LCRs are not abundant in bacteria compared to eukaryotes, and for this and other reasons, their functionality in bacteria is questioned. In order to determine the extent to which LCRs are conserved in bacterial species, and therefore perhaps functional, we decided to focus on pairs of bacterial strains. Strain pairs provide a much more focused evolutionary approach compared to what we can achieve by comparing orthologs from different species. To such end, we selected 20 pairs of completely sequenced phylogenetically diverse bacterial strains, for a total of 40 organisms (Appendix A).

For each pair of strains, we obtained all of their orthologous proteins (see Materials and Methods, Appendix A). We searched in the orthologs for homorepeats (polyX) and compositionally biased regions (CBRs). PolyX and CBRs sometimes overlap, however, by definition, polyX are shorter and CBRs include more amino acid types [24]. The number of orthologous pairs in which at least one of the orthologs has a polyX is in general slightly higher than the number of pairs with CBRs (Figure 1a).

Then, for each pair of strains, we calculated the conservation ratio of polyX, CBRs, and domains, that is, the fraction of occurrences of a feature that appear in overlapping positions of the aligned orthologs (see Materials and Methods for details). Since this conservation ratio will depend on the evolutionary distance between the strains, we also measured this parameter for reference. As a proxy for conservation between a pair of strains, we took the average bit score of the alignments of all of the pairs of orthologs found between the strains (higher scores for more similar sequences).

We also performed an analysis of the feature conservation between pairs of species using representatives from each of the 20 strains. We took one representative from each pair of strains and analyzed their 190 possible pair combinations (Appendix A). There are pairs of strains that have a level of conservation between them as low as that of most pairs of species. In fact, the species pair *Escherichia coli* K-12—*Yersinia pestis* str. CO-92 (both are Proteobacteria from order Enterobacterales) is more conserved than 8 of the 20 strain pairs. This is not too surprising if we consider that the definition of a bacterial strain is a continuum similar to that of bacterial species [16,25,26].

Domains are conserved 75% and above with little variation with sequence conservation, while CBRs and polyX only reach a similar conservation level between very close strains (Figure 1b,d). When more different (average sequence bit score < 500) strains or species are compared, CBRs are conserved in the 25–50% range, while polyX are below 25% conservation.

In the next section, we explore if there is a different conservation behavior depending on the amino acid type prevalent in the LCR.

### 3.2. LCR Conservation Patterns and Amino Acid Type

Here, we study LCR conservation considering the amino acids forming the polyX or CBR; that is, per low complexity region type, we calculate the number of conserved and non-conserved regions per amino acid per strain pair (Appendix A). 

Low count biases are a problem when dealing with LCRs in bacteria. To avoid this problem, we focus on the conservation ratio of the three most abundant polyX or CBR per strain pair. E-rich is the most frequent in the top three most abundant CBR in 14 occasions, and in 12 of these, it is more conserved than all CBRs taken together (Figure 2a). The second most frequent CBR is A-rich (in 12 strain pairs), but it is less conserved than all CBRs together in seven occasions. For homorepeats, the most abundant among the top three are polyL and polyA (in 19 and 18 pairs, respectively) (Figure 2b), with polyA most often less conserved than all polyX (five, four, and nine times above, equal, or less conserved than all, respectively). Leucine and alanine are also the two most prevalent amino acids across all strain pairs (mean amino acid frequency in the 20 pairs). Even when only the most prevalent LCRs are taken, the conservation ratio of both polyX and CBRs still correlates significantly with sequence conservation.

Next, to see if there are conservation trends for different amino acids while avoiding the problem of low numbers of LCRs, we analyzed the conservation of the polyX and CBRs of each amino acid by counting the regions of the 20 strain pairs together (Figure 2c). As observed before, polyX tend to be less conserved than CBRs. All LCRs are observed, but those with the less frequent amino acids are the rarest: polyC and polyW (one and two cases, respectively) and C- and Y-rich CBRs (17 and 24 cases, respectively). Conservation does not correlate with amino acid frequency. For example, polyA and A-rich CBRs stand out by their lower conservation regardless of their abundance. As an example, ATP-binding protein Uup from *Escherichia coli* strain K12 (UniProt: P43672) and from *Escherichia coli* O157:H7 (UniProt: A0A0H3JCY6) are 99.37% identical. Their 635 amino acid proteins differ only in four positions, two of them alanines forming a polyA in *E. coli* K12 in positions 547–550 (“AAAA” versus “PAAP”). This region is between two annotated domains, ABC transporter 2 (positions 320–546) and a C-terminal domain (positions 551–635), possibly acting as a linker. The structure of the C-terminal fragment of the *E. coli* K12 protein (amino acids 551–635) has been solved: in this structure, the region 551–563 is disordered and precedes a coiled coil domain (structure in the Protein Data Bank, PDB:2LW1) (Appendix A).

Considering CBRs with more than 100 cases (12 and 14, CBRs and polyX, respectively; solid bars in Figure 2c), the highest conservation for a CBR with more than 100 cases is observed for Q-rich regions. In eukaryotic species, they have been linked to the mediation of protein–protein interactions [27,28]. It is not clear if they perform such a function in bacteria. E-, K-, and N-rich CBRs and polyI are the next more conserved LCRs, with more than 100 cases.

### 3.3. LCRs Are More Conserved in Pathogenic Strain Pairs

Pathogenic bacteria have different functional needs than non-pathogenic bacteria. Their fitness depends on their interaction with the host [29]. Emergent host mechanisms to detect or destroy a pathogen are accompanied by a parallel evolution to overcome them (they co-evolve) [30]. Since LCRs are fast-evolving sequence features, they could facilitate fast functional evolutionary changes manifested in a host–pathogen arms race. Here, we want to evaluate whether pathogenicity plays a role in the functional conservation of LCRs in bacterial strains. Six of the strain pairs we collected are pathogenic, while thirteen of them are not (Appendix A). Interestingly, for *E. coli*, we have one pathogenic (O157:H7) and one non-pathogenic (K-12) strain [31]. This will let us compare not only between but within strains.

In our dataset, pathogenic strain pairs are in general more conserved than non-pathogenic ones (Figure 3a–c). *Fusobacterium sp*. strains, however, deviate from this pattern. Although it is generally accepted that some Fusobacterium species are pathogens [32,33]), it is possible that at least one of the strains we used in this analysis (CAG:815 and CAG:439) is non-pathogenic. No hint about the pathogenicity of these specific strains could be found in the literature.

We also observe that pathogenic strains display a higher conservation ratio of sequence features, even when comparing pairs of similarly conserved strains (mean bit score > 500, Figure 3a–c). To facilitate this comparison, and considering the tendency of the conservation ratio to follow a positive correlation to sequence similarity, we use a normalized conservation ratio as the conservation ratio divided by the mean bit score per strain pair, which we compare in Figure 3d–f.

Results show the higher conservation of LCRs in pathogenic strain pairs (Figure 3d,e), contrasting the conservation of domains in non-pathogenic strain pairs (Figure 3f). Taken together, these results suggest that LCR conservation is significantly stronger in pathogenic strains compared to non-pathogenic strains, and that this difference in conservation does not apply to domains.

### 3.4. Conservation of LCRs Is Different in the Extracellular and Outer Membrane Proteins from Pathogens

Recognition of a pathogen’s extracellular and outer membrane proteins is a crucial element of the host’s defense system. That is why these proteins in the pathogen are subject to a higher evolutionary pressure to change and not be recognized. We took advantage of the *E. coli* strain pair, which has one pathogenic and one non-pathogenic strain, to check the differential presence of LCRs in this pair of strains. We computed the presence or absence of CBRs and polyX per protein, and categorized the results per subcellular location. 

In this strain pair, extracellular proteins have the highest frequency of both CBRs and polyX, with higher frequency of polyX than CBRs in cytoplasm, cytoplasmic membrane, and periplasm, and higher frequency of CBRs than polyX in the outer membrane (Figure 4a). The differences between strains are revealing: extracellular and outer membrane proteins have a higher frequency of CBRs and polyX in the pathogenic strain. The pathogenic strain has 75% more of the predicted extracellular proteins compared to the non-pathogenic strain (84 versus 48 proteins, respectively), with its proteome only 15% larger (5062 versus 4391 proteins, respectively). There is virtually no difference in the presence ratio of CBRs or polyX in proteins located in the cytoplasm, cytoplasmic membrane, and periplasm from the two strains. This result confirms that the evolutionary pressure from the interaction with the host results in more extracellular and outer membrane proteins accumulating LCRs.

Next, we wanted to compare the conservation of LCRs in extracellular and outer membrane proteins from all pathogenic and non-pathogenic strain pairs together. Because pathogenic strains are more conserved (Figure 3a–c), the conservation ratio of LCRs is higher in pathogenic pairs for all subcellular locations (Figure 4b). To facilitate the analysis, we represent the values normalized to the levels found for the cytoplasmic proteins (Figure 4c). The results indicate that, in pathogenic strains, LCR conservation does not vary much across subcellular locations, except for the lowest value found for polyX in the outer membrane proteins. However, if we compare to pathogenic strains, the LCRs in outer membrane proteins appear much more conserved. 

Taken together, our results suggest that extracellular and outer membrane proteins might accumulate LCRs in pathogenic strains, showing increased conservation in the outer membrane proteins of pathogenic strains, and with polyX showing particularly less conservation in the extracellular proteins of pathogenic strains. 

## 4. Conclusions

To try to find functionality associated with LCRs (CBRs and homorepeats) in bacteria, we studied their conservation between strains. The feature conservation of LCRs of particular amino acid types was computed by their presence in overlapping positions of aligned sequences; global conservation of a feature was evaluated as the ratio found to be conserved. We observed that Q-rich CBRs are the most conserved, and polyA and A-rich regions are the least conserved (Figure 2). 

Interestingly, strain pathogenicity seems to play a role in LCR conservation. Both CBRs and homorepeats are generally more conserved in pathogenic strain pairs than in non-pathogenic pairs, unlike domains (Figure 3). This result is related to the subcellular location of the LCRs in the pathogenic strains: we see a higher abundance of LCRs in extracellular and outer membrane proteins. This is not surprising: pathogens use protein disorder to escape the host’s recognition system [34,35], and LCRs often produce disordered regions. However, polyX displays exceptional variability in the extracellular proteins of pathogens.

We note that due to the low number of CBRs in bacteria, we had to use relaxed cut-offs for CBR detection, particularly considering stretches of as few as four consecutive identical amino acids as a homorepeat. While it was demonstrated that such length results in structural effects related to disordered regions in polyQ [36], it is not clear if this is the case for other amino acids. For example, polyI with a length of four can adopt β strand structure within a β sheet (see structures in the Protein Data Bank, PDB:6MFV for UniProtKB:O58663, and PDB:6IGS for UniProtKB:Q5NI77). Therefore, while we have chosen both CBRs and polyX as LCRs for analysis, and their definitions have some degree of overlap, it is more likely that our observations for CBRs will be more associated with disordered regions and less dependent on amino acid type than those for polyX.

In any case, we note that CBRs are not necessarily disordered. We can show this with an example from the abundant E-rich CBRs that seem to be especially conserved between the very divergent strains of *Firmicutes bacterium* (green dot labeled “E” in Figure 2a near coordinates [200, 0.30]). The conserved E-rich region of the Lon protease (UniProt:R9LP70 /amino acids 193–269; UniProt:A0A418MLU7/192–271) is actually predicted to be a coiled coil and is shown to be part of one in one of the *E. coli* LonA solved structures (PDB:6U5Z) (Appendix A). It is more likely that non-conserved LCRs are not structured. We can show this for a non-conserved E-rich in another *F. bacterium* pair: the N-terminal region of the DNA gyrase subunit A is longer and contains an E-rich region in one ortholog (UniProt:R9LNF4/3-32) that is missing in the other (UniProt:A0A418LWL6). The structure of another ortholog in *Bacillus subtilis* confirms that the N-terminal of these proteins is disordered and of variable length (PDB:4DDQ). We presented the most general trends found in these data, but there are many possible focused analyses. We provide Appendix A with all the data necessary to assess each of these cases in future individual analyses that should take advantage of the increased availability of solved structures. Our results clearly show that the conservation of LCRs in bacterial proteins depends on their subcellular location and strain pathogenicity. We believe that this indicates LCR association with bacterial functions related to host–pathogen interactions, constituting a step towards the functional acknowledgment of LCRs in bacteria.

## Figures and Tables

**Figure 1 genes-12-00451-f001:**
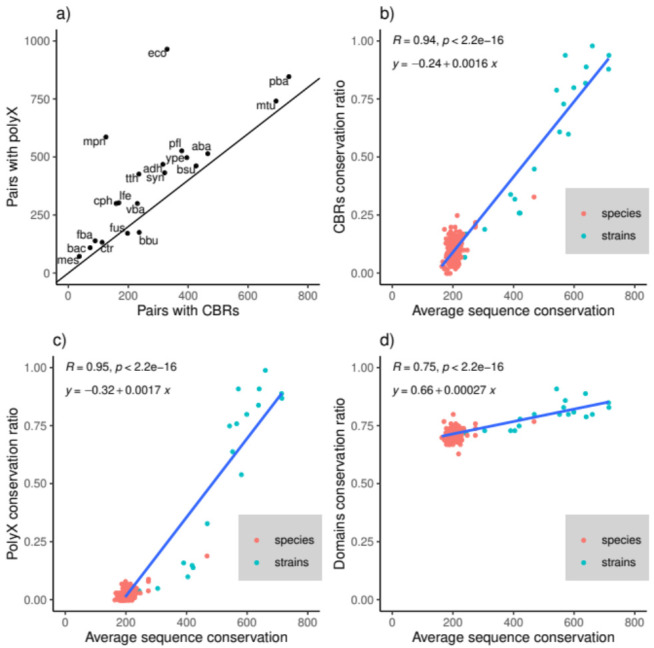
(**a**) Number of orthologous pairs between strains with polyX versus the number of pairs with compositionally biased regions (CBRs); in black, line x = y; (**b**) CBRs; (**c**) polyX, and (**d**) domain conservation ratio versus average full-sequence conservation in orthologs from pairs of strains (or species). Sequence conservation is measured as the average bit score of all orthologous pairs. The regression line is plotted in blue.

**Figure 2 genes-12-00451-f002:**
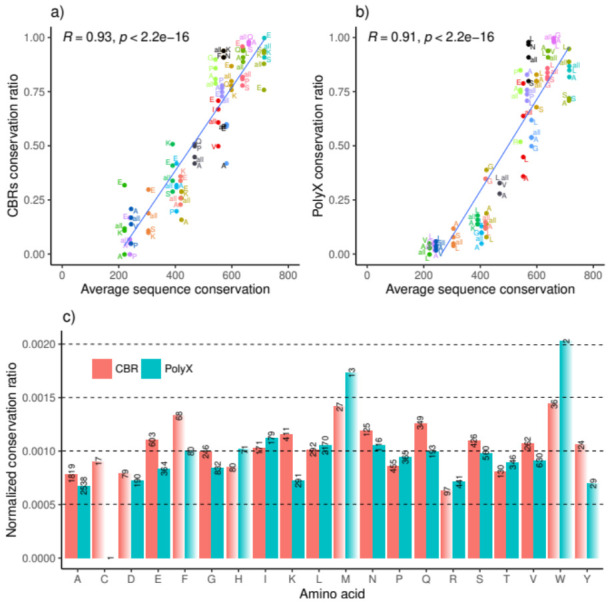
(**a**) CBR conservation ratio per bacterial strain pair compared to the average conservation (in bit score) of all orthologous pairs for the top three most abundant CBRs per strain pair; (**b**) polyX conservation ratio per bacterial strain pair compared to average conservation (in bit score) of all orthologous pairs for the top three most abundant polyX per strain pair; (**c**) conservation ratio for CBRs and polyX (of each amino acid type) normalized by the average conservation of the strain pair; bars with less than 100 cases are blurred. Dots for each strain pair are represented with the same color in (**a**,**b**) and align vertically since they have the same x-axis value; the value “all” (displayed in Figure 1b,c) is included for comparison.

**Figure 3 genes-12-00451-f003:**
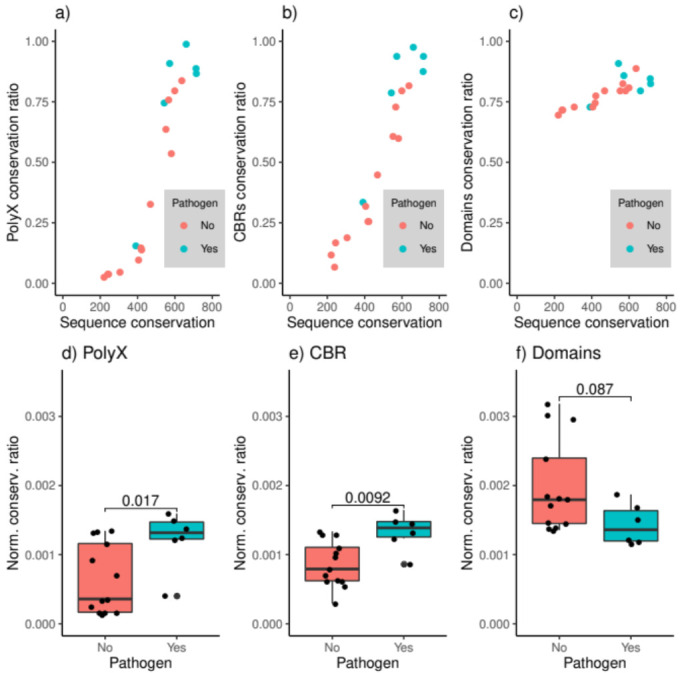
(**a**–**c**) PolyX, CBRs, and domain conservation ratio compared to sequence conservation (mean bit score) for pathogenic and non-pathogenic strain pairs; (**d**–**f**) polyX, CBRs, and domain conservation ratio normalized by the sequence conservation for pathogenic and non-pathogenic strain pairs. Non-parametric Mann–Whitney U statistical tests were performed to compare the distributions.

**Figure 4 genes-12-00451-f004:**
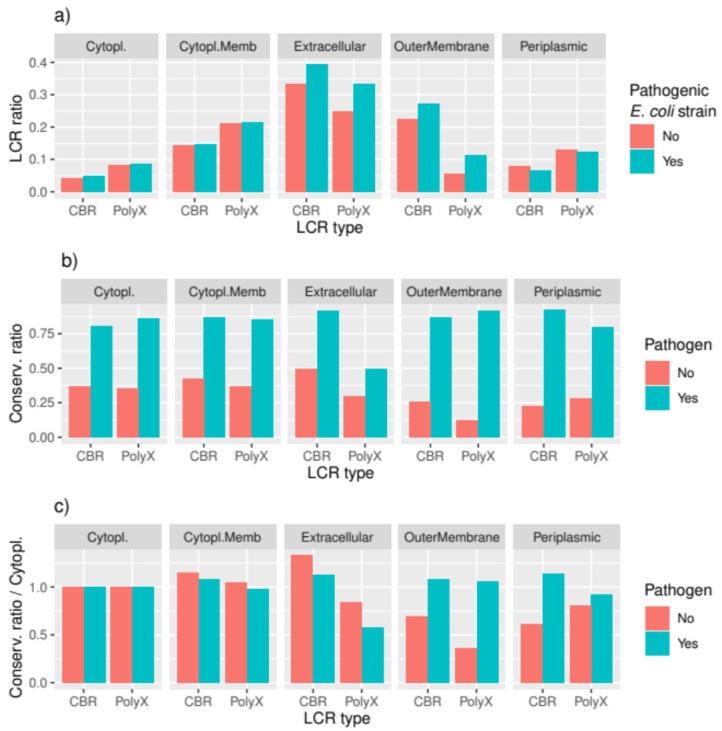
(**a**) CBR and polyX abundance (number of proteins with at least one region versus total number of proteins) per subcellular location and *E. coli* strain; (**b**) conservation ratio for CBR and polyX regions per subcellular location, taking all pathogenic and non-pathogenic strains together; (**c**) conservation ratio for CBR and polyX regions per subcellular location, normalized by values obtained for cytoplasmic proteins. LCR: low complexity region.

## Data Availability

The data presented in this study are available as Appendix A.

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
