# Peer review of "The Conservation of Low Complexity Regions in Bacterial Proteins Depends on the Pathogenicity of the Strain and Subcellular Location of the Protein"

_genes, 2021, doi:10.3390/genes12030451_

Round 1
Reviewer 1 Report
This manuscript conducted an extensive search of low complexity regions (LCRs) to show the high conservation degree in pathogens. The experiments are well designed, and the results are interesting. However, I would like the authors to consider the following points.
- The description of the methods is not enough.
I cannot understand the definition of conservation. The authors took overlap in an alignment for the detections of the conservation. However, I cannot understand “overlap”. When LCR in 10 residues exists, does the only 1 residue overlap in the alignment meet the threshold ? I think that 1 residue overlap for 10 residue LCR is not “conserved”.
When the lengths of LCRs differ in a pair of orthologs, how the authors deal with this ? When strain 1 has poly-X of 5 residues and strain 2 has ploy-X of 10 residues, what happens ?
When the alignment has in/del in a LCR, how the authors deal with this for conservation ?
The authors measured the distance between strains or species by “average bit score of the alignments of their orthologs”. Does it mean that the distance is measured by taking the average bit-score over all the orthologs found in a pair of strains ?
Is the bit-score here the score summing a score for a pair of aligned resides over an alignment ? If so, which program and scoring matrix were used ?
When highly conserved two domains exists, the region in between these domains can be aligned even if the sequences are not similar. When CBRs are diverged in a pair of strains and this CBR locates in such region, the alignment program aligns these regions with no similarity. In this case, these CBRs must be overlapped. I wonder the similarity of the aligned CBRs needed to be considered ?
- I would like the authors to show the trends of LCRs.
I think there are several variations in CBR and ploy-X. I would like to know some trends of them, for example, length distributions of CBR and ploy-X (break down into each residue), how abundant of X-rich LCRs, etc.
What is the definition of X-rich CBR ?
I could not understand Fig. 2 a) and b). I could not follow the pairs of colors and residues (20 colors are too many to recognize). Then, I could not follow these figures.
- This manuscript shows the general trend of the conservation of LCR, which is fine. However, some detailed information can help the readers understand the results. For example, I tried to figure out the LCR structure in PDB:6u5z (line292), but I could not find the LCR in E. coli sequence. Then, some alignments or domain diagrams are helpful, for example, for lines 178-186, and examples in Conclusions.
UniProt provides functional annotations. Could the authors find functional annotations for LCRs ?
- Minor points
The legends for Supplemental Figs and Tables are needed. I could not understand these Figs and Tables without legends.
Line 117 shows what ployX and CBR stand for. However, the first appearances of them are in line 93.
Author Response
This manuscript conducted an extensive search of low complexity regions (LCRs) to show the high conservation degree in pathogens. The experiments are well designed, and the results are interesting. However, I would like the authors to consider the following points.
1) The description of the methods is not enough.
I cannot understand the definition of conservation. The authors took overlap in an alignment for the detections of the conservation. However, I cannot understand “overlap”. When LCR in 10 residues exists, does the only 1 residue overlap in the alignment meet the threshold ? I think that 1 residue overlap for 10 residue LCR is not “conserved”.
We have included a clarification about this in lines 100-101.
When the lengths of LCRs differ in a pair of orthologs, how the authors deal with this ? When strain 1 has poly-X of 5 residues and strain 2 has ploy-X of 10 residues, what happens ?
The reply to the previous comment (clarification in lines 100-101) should serve to reply to this comment too. In this case, if the complete shortest polyX (5 residues) overlaps with the longest polyX (10 residues), it would be considered as conserved (“at least 50% of the longest region overlaps with the predicted feature in the ortholog”).
When the alignment has in/del in a LCR, how the authors deal with this for conservation ?
We refer throughout the manuscript to feature conservation, not residue conservation: a feature is either conserved (present in overlapping positions of both orthologs) or not conserved. Taking into account the length of the overlap in the alignment between both LCR features is enough to cover this.
The authors measured the distance between strains or species by “average bit score of the alignments of their orthologs”. Does it mean that the distance is measured by taking the average bit-score over all the orthologs found in a pair of strains ?
Yes. We have rephrased it in the manuscript, in lines 135-136. In any case, it was already stated in the original manuscript in lines 87-89.
Is the bit-score here the score summing a score for a pair of aligned resides over an alignment ? If so, which program and scoring matrix were used ?
Yes. We obtained the bit scores from the program ProteinOrtho v6.0.23 with default parameters (already stated in the original manuscript, lines 85-86). We have included an additional clarification on this in lines 86-87.
When highly conserved two domains exists, the region in between these domains can be aligned even if the sequences are not similar. When CBRs are diverged in a pair of strains and this CBR locates in such region, the alignment program aligns these regions with no similarity. In this case, these CBRs must be overlapped. I wonder the similarity of the aligned CBRs needed to be considered ?
Please, note that in our analysis of CBRs we consider their amino acid type, therefore, the type of amino acid enriched will be the same in both orthologs, which is a comparison that is stricter than mere overlap.
2) I would like the authors to show the trends of LCRs.
I think there are several variations in CBR and ploy-X. I would like to know some trends of them, for example, length distributions of CBR and ploy-X (break down into each residue), how abundant of X-rich LCRs, etc.
The abundance of CBR and polyX regions per strain was already included in the original submission of the manuscript, in Supplementary Figure S2. We now include in this figure two more panels, featuring the length distribution per type of CBR and polyX. We have included a comment on this modification in the manuscript, in line 317.
What is the definition of X-rich CBR ?
The definition of CBRs was already included in the original submission of the manuscript, in lines 37-38, with a second mention in lines 120-121. Appropriate references were also included.
I could not understand Fig. 2 a) and b). I could not follow the pairs of colors and residues (20 colors are too many to recognize). Then, I could not follow these figures.
Figures 2a and 2b are a decomposition of Figures 1b and 1c, featuring the top-3 most abundant CBR or polyX per strain pair; we have included a small comment on this in Figure 2 legend (line 175). Coloring should not be a problem, as the 4 points per strain pair align vertically (they have the same x-axis value), and we have made sure that no similar colors have proximal x-values.
3) This manuscript shows the general trend of the conservation of LCR, which is fine. However, some detailed information can help the readers understand the results. For example, I tried to figure out the LCR structure in PDB:6u5z (line292), but I could not find the LCR in E. coli sequence. Then, some alignments or domain diagrams are helpful, for example, for lines 178-186, and examples in Conclusions.
We illustrate these cases with new Supplementary Figures S4 and S5.
UniProt provides functional annotations. Could the authors find functional annotations for LCRs ?
Functional annotations of LCRs in bacteria were discussed in [7]. We now include a comment about this in the Introduction (line 50).
4) Minor points
The legends for Supplemental Figs and Tables are needed. I could not understand these Figs and Tables without legends.
Legends were already included in the original submission of the manuscript, in section “Supplementary Materials” (lines 315-326), right after section “Conclusions”. This is as per the journal format requirements.
Line 117 shows what ployX and CBR stand for. However, the first appearances of them are in line 93.
These two terms were already explained in the Abstract, lines 18-19. Also, in the Introduction (line 36).
Reviewer 2 Report
The manuscript submitted by Mier and Andrade analyses the conservation degree of low complexity regions and of compositionally biased regions in bacterial proteomes, with the aim to get some information about their biochemical/biological role. If the description and analysis of the conservation degree of these peptide moieties is remarkable, basically no information was obtained on the biochemical/biological side. As it would have been expected, in my opinion. For this is probably better to focus the attention of a specific protein – its structure, its binding partners, its post-translational modifications etc. etc.
Nevertheless, I think that the paper is interesting and deserve publication, with very minor modifications.
1) It is not really clear how the overlap of a feature (polyX or CBR) in the sequence alignment is defined: just one alignment position, at least N alignment positions, at least X% etc. Would it be possible for the Authors to indicate it explicitly?
2) Figure 1. The orange spots are extremely clustered – because of the axes scaling – and it is impossible to visualize any correlation between X and Y.
3) In section 3.3 the Authors should indicate if these bacteria a pathogenic for humans (or for other organisms).
Author Response
The manuscript submitted by Mier and Andrade analyses the conservation degree of low complexity regions and of compositionally biased regions in bacterial proteomes, with the aim to get some information about their biochemical/biological role. If the description and analysis of the conservation degree of these peptide moieties is remarkable, basically no information was obtained on the biochemical/biological side. As it would have been expected, in my opinion. For this is probably better to focus the attention of a specific protein – its structure, its binding partners, its post-translational modifications etc. etc.
Nevertheless, I think that the paper is interesting and deserve publication, with very minor modifications.
1) It is not really clear how the overlap of a feature (polyX or CBR) in the sequence alignment is defined: just one alignment position, at least N alignment positions, at least X% etc. Would it be possible for the Authors to indicate it explicitly?
We have included a clarification about this in lines 100-101.
2) Figure 1. The orange spots are extremely clustered – because of the axes scaling – and it is impossible to visualize any correlation between X and Y.
We have tried to change this by reducing the point size, but they are still ~190 points and it would be very difficult to visualize them all, given their very similar x- and y- values. To visualize the correlation, as stated by the reviewer, we included the R coefficient and the correlation equation (also plotted with the blue line) in each panel. We now include a comment on the regression line in lines 128-129. In any case, these orange points refer to species pairs, which act as the “background” conservation in this representation; they are not the pairs in study.
3) In section 3.3 the Authors should indicate if these bacteria a pathogenic for humans (or for other organisms).
We do not think there is any difference in the functional implications of the LCR with respect to the bacterial host, but we do acknowledge that it is a good point that may be worth of further research. We have included this information in Supplementary File S1.
Round 2
Reviewer 1 Report
The answers are mostly acceptable, but I would like to the authors to add the followings.
I understand that this manuscript analyzed the conservation of “features” and did not conservation of sequence. However, some reader may misunderstand it like me because the term conservation reminds “sequence conservation”. Then, it is better to emphasize of this point.
I would like to know how rich X residue in X-rich CBR. There must be cutoff to found X-rich CBR.
The definition of X-rich CBR should be denoted, like “more than y% X residues in a CBR is regarded as X-rich CBR”.
Author Response
The answers are mostly acceptable, but I would like to the authors to add the followings.
1) I understand that this manuscript analyzed the conservation of “features” and did not conservation of sequence. However, some reader may misunderstand it like me because the term conservation reminds “sequence conservation”. Then, it is better to emphasize of this point.
We thank the referee for making this point in need of clarification. We have added an explicit comment about this at the end of the Introduction (lines 72-77) and at the start of the Conclusions (lines 283-286).
2) I would like to know how rich X residue in X-rich CBR. There must be cutoff to found X-rich CBR. The definition of X-rich CBR should be denoted, like “more than y% X residues in a CBR is regarded as X-rich CBR”.
We have included a clarification on this in Materials and Methods, lines 102-105.